# GiOPARK Project: The Genetic Study of Parkinson’s Disease in the Croatian Population

**DOI:** 10.3390/genes15020255

**Published:** 2024-02-19

**Authors:** Valentino Rački, Gaber Bergant, Eliša Papić, Anja Kovanda, Mario Hero, Gloria Rožmarić, Nada Starčević Čizmarević, Smiljana Ristić, Saša Ostojić, Miljenko Kapović, Aleš Maver, Borut Peterlin, Vladimira Vuletić

**Affiliations:** 1Department of Neurology, Faculty of Medicine, University of Rijeka, 51000 Rijeka, Croatia; valentino.racki@uniri.hr (V.R.); elisa.papic@uniri.hr (E.P.); mario.hero@student.uniri.hr (M.H.); gloria.rozmaric@student.uniri.hr (G.R.); 2Department of Neurology, Clinical Hospital Center Rijeka, 51000 Rijeka, Croatia; 3Clinical Institute of Genomic Medicine, University Medical Centre Ljubljana, 1000 Ljubljana, Slovenia; gaber.bergant@kclj.si (G.B.); anja.kovanda@kclj.si (A.K.); ales.maver@kclj.si (A.M.); borut.peterlin@kclj.si (B.P.); 4Department of Medical Biology and Genetics, Faculty of Medicine, University of Rijeka, 51000 Rijeka, Croatia; nadasc@medri.uniri.hr (N.S.Č.); smiljana.ristic@medri.uniri.hr (S.R.); sasa.ostojic@medri.uniri.hr (S.O.); miljenko.kapovic@medri.uniri.hr (M.K.)

**Keywords:** Parkinson’s disease, genetic testing, whole-exome sequencing

## Abstract

Parkinson’s disease is a neurological disorder that affects motor function, autonomic functions, and cognitive abilities. It is likely that both genetic and environmental factors, along with age, contribute to the cause. However, there is no comprehensive guideline for genetic testing for Parkinson’s disease, and more research is needed to understand genetic variations in different populations. There has been no research on the genetic background of Parkinson’s disease in Croatia so far. Therefore, with the GiOPARK project, we aimed to investigate the genetic variants responsible for Parkinson’s disease in 153 Croatian patients with early onset, familial onset, and sporadic late-onset using whole-exome sequencing, along with multiplex ligation-dependent probe amplification and Sanger sequencing in select patients. We found causative variants in 7.84% of the patients, with GBA being the most common gene (4.58%), followed by PRKN (1.96%), ITM2B (0.65%), and MAPT (0.65%). Moreover, variants of uncertain significance were identified in 26.14% of the patients. The causative variants were found in all three subgroups, indicating that genetic factors play a role in all the analyzed Parkinson’s disease subtypes. This study emphasizes the need for more inclusive research and improved guidelines to better understand the genetic basis of Parkinson’s disease and facilitate more effective clinical management.

## 1. Introduction

Parkinson’s disease (PD) is a common neurological disorder, accounting for a 1% worldwide prevalence in people above the age of 65 and manifesting as progressive motor dysfunctions, sometimes also accompanied by dysregulated autonomic functions and cognitive decline [1,2]. A characteristic feature of PD is the loss of dopaminergic neurons in the substantia nigra and the accumulation of Lewy bodies [3], which leads to the shortage of dopamine and the development of motor symptoms such as tremor, bradykinesia, rigidity, and postural instability. Non-motor symptoms such as depression, apathy, visual hallucinations, and dementia also occur, often preceding the motor symptoms [4].

Parkinson’s disease is believed to be caused by a combination of genetic and environmental factors, according to existing research. The typical age for the onset of symptoms is around 60 years, with onset before or after the age of 50 being considered as early-onset or late-onset PD, respectively [1,5,6,7]. The diagnosis of PD is strictly based on clinical criteria [8], although recent papers have called for a bridging between clinical and diagnostic procedures [9]. Older diagnostic criteria, such as the United Kingdom Parkinson’s Disease Society Brain Bank Clinical Diagnostic Criteria (UKPDSBB), list a first-degree relative with PD as an exclusion criterion [10], which does not agree with known advancements in the field.

Approximately 50 genes have been identified as causing autosomal dominant, autosomal recessive, and in rare cases, X-linked Mendelian forms of PD; however, many proposed genes have not been ultimately confirmed through independent validation [2,11]. Nevertheless, there is increasing evidence that genetic background can influence the risk, onset, and progression of PD [12]. However, the latest recommendations for genetic testing are from 2013 and come from the EFNS/MDS criteria. These criteria suggest that genetic testing be conducted on an individual basis, taking into account family history and the age of onset, but only for selected genes (*SNCA*, *LRRK2*, *GBA*, *PINK1*, *DJ-1*, *ATP13A2*, *PLA2G6*, and *FBXO7*), as next-generation sequencing was not widely available at the time [13]. There are still no standard genetic panels or recent guidelines that can be applied to all PD patients [14].

The whole-exome sequencing (WES) approach has recently enabled the simultaneous analysis of many genes or gene panels, potentially improving the diagnostic yield of genetic testing [15]. Despite the advancements in genetic testing for PD, it is not commonly practiced. According to a survey conducted in 2019, 41% of experienced PD physicians did not perform genetic testing in the past year. Additionally, more than 80% referred fewer than 11 patients for genetic testing during that same period [16]. Most patients referred to genetic testing in PD have early onset (EO) or familial onset (FO). Therefore, information on the expected diagnostic yield of genetic testing is important for informing clinical decisions. To address this issue, we recently published a collaborative next-generation sequencing (NGS) genetic study in EO and FO PD patients from the Serbian, Slovenian, and Croatian populations, the first South Slavic population study [17].

Furthermore, there is a lack of diversity in current research on the genetic basis of PD, as most studies involve patients from Western Europe, China, and the United States of America [14]. Recently, the data of a global initiative involving a cohort of 3888 patients who had positive results from genetic testing were published [18]. The most significant numbers were *GBA* (38.5%) and *LRRK2* (41%) variants, but variants were also present in *PRKN* (13.5%), *PINK1* (2.3%), *DJ-1* (0.1%), *VPS35* (0.7%), and *SNCA* (3.6%) genes. Although this was a global initiative, the majority of the patients were from Germany (18.1%), Israel (9.5%), and Spain (8.6%), i.e., 89.6% of the white population, which points to differences in the availability of genetic testing. No patients from our country and neighboring countries were in the mentioned cohort [18]. Furthermore, a large-scale study is currently underway in the United States, which aims to determine the frequency of variants in the seven most common genes associated with Parkinson’s disease: *GBA*, *LRRK2*, *PRKN*, *SNCA*, *PINK1*, *PARK7*, and *VPS35* [19]. It is planned to include 15,000 subjects; currently, 7500 have been sequenced, of which 14% have pathogenic mutations. The presented data extracted from 1959 individuals revealed that 290 (14.8%) had reportable variants, mainly in the *GBA* gene (9.2%), as well as *LRRK* (2.9%) and *PRKN* (2.6%), most of which were heterozygous variants [20]. The limitation of the mentioned study is its focus on the seven most common genes, although there is always the possibility of expanding the analysis. Finally, there is a project underway involving over 136 cohorts in 57 locations worldwide [21]. Future results will include data from an expected 161,905 patients, and so far, 14,902 patients (9.20% of the planned samples) have been sequenced. It is evident that soon, there will be an outstanding resource for studying the genetics of Parkinson’s disease available to all.

There have been no studies on the genetic background of PD in Croatian patients thus far. Due to this, we started the GiOPARK project (“The Epidemiology of Parkinson’s Disease in Croatia and the Influence of Genetic Factors and Microbiota on the Progression and Treatment Outcomes of the Disease”) [22], with which we aimed to assess not only the genetic background but also the influence of genetic and microbial factors on disease progression in an ongoing longitudinal study. With this phase of the project, our aim was to assess the genetic background of Croatian PD patients using ES and MLPA and, at the same time, to assess the diagnostic yield in EO, FO, and sporadic onset (SO) groups.

## 2. Materials and Methods

Our study consisted of 153 patients with PD who were referred for routine genetic testing at the Clinical Hospital Centre Rijeka and through patient outreach programs in collaboration with the Croatian Parkinson’s Disease patient association “Parkinson i mi,” from January 2020 to May 2023. During their clinical appointment, the patients provided informed consent for genetic testing. The inclusion criteria for the study were a confirmed clinical diagnosis of PD, based on UKPDSBB clinical diagnostic criteria [10]. Patients were divided into three groups based on the disease presentation: early onset (<50 years of age and no family history), familial onset (positive family history), and sporadic late onset (>50 years of age and no family history). The exclusion criteria for the study included inconsistent clinical presentation, Parkinson-plus syndromes, or insufficient clinical information.

### 2.1. Gene Panel Analysis

Exome sequencing was carried out through a collaboration between the Department of Medical Biology and Genetics, Faculty of Medicine, University of Rijeka, (Rijeka, Croatia) and the Clinical Institute of Genomic Medicine, University Medical Center Ljubljana (Ljubljana, Slovenia). Standardized protocols were used during the process. Exome sequencing capture was performed using different versions of the Twist Library Preparation Kit (Twist Bioscience, San Francisco, CA, USA). Illumina sequencing platforms were used for sequencing in either 2 × 100 or 2 × 150 paired-end sequencing mode. The sequencing data analysis, including variant annotation, storage, and extended data analysis, was performed as previously described [23,24,25,26,27].

Variant interpretation was carried out during the duration of the study, as previously reported [28]. To account for method improvements during the duration of the study, we reanalyzed the raw data from all the samples using the most recent software and annotation databases to reduce variability in the analysis procedure. We analyzed a total of 162 genes, with the information on our two panels included in Appendix A.

The identified variants were classified according to the ACMG and AMP 2015 joint consensus recommendation [29]. The evidence support level was additionally weighted according to the ACGS recommendations where applicable [30]. Pathogenic and likely pathogenic (P/LP) variants in patients consistent with the inheritance model of individual gene disorders were considered clinically relevant.

### 2.2. Multiplex Ligation-Dependent Probe Amplification Analysis

To detect deletions or duplications in *SNCA*, *PRKN*, *UCHL1*, *PINK1*, *PARK7*, *ATP13A2*, *LRRK2*, and *GCH1* genes in patients with early-onset Parkinson’s disease, we used the semi-quantitative multiplex ligation-dependent probe amplification (MLPA) SALSA MLPA Probemixes P051 and P052. These assays were obtained from MRC Holland, Amsterdam, The Netherlands.

### 2.3. Sanger Sequencing

To address the challenge of the high homology between the *GBAP1* and *GBA* genes, we carried out a verification process for variants detected in homologous gene regions. We focused on verifying any pathogenic variants flagged as poor quality by the variant caller through Sanger sequencing. Additionally, we performed a manual raw sequencing data review in all negative patients to determine the existence of two commonly detected pathogenic variants in regions of high homology, c.1448T>C and c.586A>C, which could escape the detection by the variant caller.

### 2.4. Statistical Analyses

Statistical analysis was performed in JASP (Amsterdam, The Netherlands). Tests were selected based on data distribution. A binomial test was used for sex distribution. An ANOVA test with Tukey post hoc analysis was used for the comparison of time to diagnosis between the groups. The chi-square test was used to assess the differences in the incidence of causative variants between the groups. Statistical significance was set at *p* < 0.05.

## 3. Results

A total of 153 patients participated in this research, 42 of whom had EO (*n* = 42, 27.45%), 40 had FO (*n* = 40, 26.14%), and 71 had SO (*n* = 71, 46.41%). The study cohort comprised 89 men (58.17%) and 64 women (41.18%) patients, with the difference being significant (*p* = 0.0084). The FO patient group did not have statistically significant differences between sexes (*n* = 20 M, 50%, *p* = 0.1253), which was observed in both SO (*n* = 44 M, 61.97%, *p* = 0.0124) and EO (*n* = 25 M, 59.25%, *p* = 0.05). We also analyzed the age at which the patients noticed the first symptoms. In the case of the entire cohort, the age of onset was 56.49 ± 12.61 years, while as expected, it was less for EO (42.07 ± 7.78), average for FO (57.55 ± 12.03), and more in the case of SO (63.8 ± 7.11). There were also significant differences in the time of disease diagnosis in the cohort between all groups (F = 25.229, *p* < 0.00001). In the entire cohort, this was an average of 1.52 ± 1.01 years, while it was significantly longer in the case of EO (2.97 ± 0.83, Q = 7.19, *p* < 0.05) and significantly shorter in the case of FO (1.04 ± 0.34, Q = 5.18, *p* < 0.05). In the case of SO, the time to diagnosis was similar to the entire group (1.68 ± 0.98, Q = 1.00, *p* = 0.892) (Table 1). The limitation of these data is that, for many patients, the data are self-reported and not verified in the hospital’s systems, especially for patients not primarily treated at our center.

All patients underwent whole-exome sequencing according to the previously described procedure, with the mean on-target coverage of 217× achieved across the samples included in our study. The obtained variants were analyzed according to ACMG and ACGS criteria [29,30]. Causative pathogenic and likely pathogenic variants were found in 13 patients (8.50%) (Table 2). Most of the causative variants were found in the *GBA* gene (nine variants in seven patients) at locations c.1226A>G (*n* = 3), c.1289C>T (*n* = 2), c.1342G>C (*n* = 2), c.1448T>C (*n* = 1), and c.882 T > G (*n* = 1). This was followed by the *PRKN* gene, with two causative homozygous variants within one family c.823C>T (*n* = 2) and one heterozygous variant, and the *ITM2B*, *SNCA*, and *MAPT* genes, with one causative pathogenic mutation each at locations c.564+1G>T (*ITM2B*), c.44T>C (*SNCA*), and c.2092G>A (*MAPT*) (Table 2).

When analyzing each group individually, six EO patients had pathogenic and likely pathogenic variants (*GBA n* = 5, c.1289C>T, c.882 T > G, c.1226A > G, c.1342G > C, and c.1448T>C; *PRKN n* = 3, 2× c.823C>T homozygous, and 1× c.823C>T heterozygous). Four FO patients had causative variants (*GBA n* = 3, 2× c.1226A>G; 2× c.586A>C; *ITM2B n* = 1, c.564+1G>T; and *SNCA n* = 1, c.44T>C). Finally, three causative variants were found in the SO as well (*GBA* c.1289C>T, c.1342G>C; *MAPT* c.2092G>A) (Table 2). Comparing the groups, we can see that the causative variants were most found in the EO group (13.95%), while variants in the FO (10%) and especially the SO group (4.23%) were less frequent. However, the difference between all three groups was not statistically significant (*X*^2^ (2, *N* = 153) = 3.59, *p* = 0.1659.).

Furthermore, we detected two variants in the *GBA* gene, c.1448T>C and c.586A>C, in regions of high homology in multiple patients. Variant calling in this region is not reliable using short-read sequencing [31]. For this reason, variants underwent confirmation using Sanger sequencing, and ultimately, only one c.1448T>C variant was confirmed in the tested patients. The c.1448T>C variant was detected in three cases, with Sanger sequencing confirming its presence in one of these cases. Initially, the c.586A>C variant was identified in seven cases; however, it was not confirmed in any of them through Sanger sequencing. 

During the analysis of the sequenced variants, many variants were found for which the clinical significance in patients is unclear, i.e., variants of uncertain significance (VUSs). Of the 153 patients, 39 had at least one VUS (*n* = 40, 25.49%). When we look at the division by onset, VUSs were found in 12 patients in patients with early onset (*n* = 12, 28.57%), 6 patients with familial onset (*n* = 5, 12.5%), and 23 patients in sporadic form (*n* = 23, 30.98%) (Table 1). The differences between the groups were not statistically significant (*X*^2^ (2, *N* = 153) = 2.63, *p* = 0.26). A total of 49 VUSs were found in the following genes: *GBA* (*n* = 1), *ITM2B* (*n* = 1), *LRRK2* (*n* = 2), *ATP13A2* (*n* = 6), *SNCA* (*n* = 1), *VPS35* (*n* = 1), *CHCHD2* (*n* = 1), *DNAJC13* (*n* = 2), *EIF4G1* (*n* = 3), *GIGYF2* (*n* = 1), *CCDC88CC* (*n* = 2), *DNMT1* (*n* = 1), *GRN* (*n* = 1), *KCNC3* (*n* = 2), *ERBB4* (*n* = 2), *GCDH* (*n* = 1), *HRPNPA1* (*n* = 1), *KIF5A* (*n* = 2), *NOTCH3* (*n* = 1), *NPC1* (*n* = 1), *NR4A2* (*n* = 1), *OPTN* (*n* = 1), *PSEN1* (*n* = 3), *PDGFRB* (*n* = 1), *SETX* (*n* = 2), *SORL1* (*n* = 1), *TBK1* (*n* = 1), *THAP1* (*n* = 1), and *VPS13C* (*n* = 4) (Appendix A).

## 4. Discussion

In this study, we performed whole-exome sequencing in 153 patients, with causative variants found in 13 (8.50%). By far, the most common causative variants were found in the *GBA* gene, found in 7 of the 13 patients (53.84%), followed by the *PRKN* gene (23.08%) and the *ITM2B*, *SNCA*, and *MAPT* genes (7.70% each). The study cohort included more men than women, which is expected due to the higher risk of developing PD in males [32]. The only exception was the FO group, although this could be due to the small sample size. In most populations, there is still a lack of genetic studies that include patients in all possible onset types of PD and often focus on patients with a positive family history of PD or an early onset of the disease. A study has been published involving several patients from this cohort as part of a larger group of patients with early and familial onset from Croatia, Slovenia, and Serbia, where *GBA* was the most common causative genetic factor for the development of Parkinson’s disease, with a higher percentage of patients with causative variants, possibly because sporadic patients were not included [17]. Similar results are observed in neighboring Italy, in an isolated population in Sardinia, where *GBA* and *PRKN* variants were the most common but in a smaller percentage than in our study (4.4%) [33]. However, patients were not stratified in relation to onset. Furthermore, in an Italian population, a significantly higher rare variant burden in the *GBA* gene was found in PD patients (20.5%) than in controls (7.2%). These findings contributed to the increased polygenic risk scores, even in sporadic onset patients [34].

Trinh et al. performed whole-exome sequencing in a German cohort of 50 patients with early-onset disease. They found seven causative pathogenic mutations: three in *GBA*, three in *PRKN*, and one in the *PLA2G6* gene [35]. Their yield of 14% was similar to ours in the early-onset subgroup, along with the same two genes (*GBA* and *PRKN*). In a mixed German–Czech population of 80 patients with early onset, 11.25% of pathogenic variants were found, more than half of which were in the *GBA* gene, while variants in the *LRRK2* and *PRKN* genes were present to a lesser extent [36]. In the Spanish population of patients with the early onset of the disease, a significant percentage of pathogenic and probably pathogenic variants (22.22%) were found, most of which were in the *PRKN*, *LRRK2*, *GBA*, and *ATP13A2* genes [37]. The main risk factors for the development of PD in a Finnish population were variants in the *GBA* gene (p.N370S and p.L444P) and to a lesser extent in the *LRRK2* gene and *POLG1* CAG nucleotide repeats [38]. In other European populations, *GBA* and *PRKN* mutations were also the most common causes, such as in Belgian [39] and mixed Dutch–British populations [40]. In an isolated population on the Faroe Islands, only one pathogenic mutation in the *LRRK2* gene was found in 91 patients of all types of presentation. In contrast, several variants of uncertain significance were found in the *LRRK2*, *ATP13A2*, and *DNAJC13* genes [41]. Such differences indicate the importance of studying different populations because there are significant changes, especially in isolated populations.

Considering patient populations in other continents, those in Asia and North America were most often analyzed. As in European populations, the most significant variants were in the *GBA* [42] and *PRKN* genes [43]. In addition to the above, mutations in the *LRRK2*, *SNCA*, *ATP13A2*, *TMEM230*, and *FBXO7* genes were significant in extensive population studies of the early onset of the disease [43]. The overall percentage of pathogenic mutations found was 7.88%, slightly lower than in our study, even though it was focused on earlier disease onset. Similar findings were revealed in other Asian populations, but there were multiple variants in autosomal recessive genes, such as *PRKN*, *PINK1*, and *PLA2G6* (9.3%) [44]. In addition, frequent variants were present in the *GBA* and *LRRK2* genes, which were also present in the other mentioned populations in this discussion [44]. However, there are significant differences in rarer variants, where variants found in new genes are not equal between European and Asian populations [45]. In a Central Asian study of an early-onset Parkinson’s disease population in Kazakhstan, pathogenic mutations were found in only 6% of cases and only in the *LRRK2* and *PRKN* genes, confirming regional differences between populations [46]. Another study on a heterogeneous population of 203 patients and all types of presentation showed a high yield of pathological variants of 20.19%, mainly in the *GBA* (10%) and *LRRK2* (3%) genes [47]. Slightly lower but significant percentages were found in American patients of European origin, where the percentage of *GBA* variants was 7.1%, and *SNCA*, *LRRK2*, and *PRKN* variants were 2% [48]. Recently, a genome-wide association study in a population of African descent discovered a new risk variant in the *GBA* gene, which was not present in non-African populations, signifying the importance of research and testing inclusiveness [49]. Furthermore, in targeted studies that focused on more common genes, variants were present in the *PRKN* and *GBA* genes but not in other expected genes for monogenic diseases [50,51].

We also included a heterozygous pathogenic *PRKN* variant as causative, even though there is still a debate on whether that is the case. It is still not conclusive whether they pose a risk for PD due to opposing studies [52,53,54,55,56]. However, it is essential to note that functional studies show impaired mitochondrial function even in the presence of heterozygous variants, indicating that there might be a biological basis for the disease [57]. Also, we found two variants in the *GBA* gene, c.1448T>C and c.586A>C, in regions of high homology in multiple patients. Variant calling in this region is not reliable using short-read sequencing. For this reason, the variants underwent confirmation using Sanger sequencing. The c.586A>C variant was identified as a low-quality variant within our pipeline in seven cases. However, Sanger sequencing did not allow us to confirm it in any of them. Similarly, the c.1448T>C variant was also identified as low-quality and confirmed in one of three patients. Notably, the highly homologous pseudogene *GBAP1* can make the NGS analysis of *GBA* difficult [31]. Even though long-read NGS can help mitigate this issue, it is still limited in recognizing recombinant alleles [58]. A recent study presented an NGS pipeline for the analysis of the *GBA* gene, which also highlights the importance of copy number variants in the gene [59]. These alterations could play a role in the varying penetrance for *GBA*, which can be seen in our study if we look at the ages of onset for the variants. In general, there is much to be done to fully understand how even the more common causative genes, such as *GBA*, *LRRK2*, and *PRKN*, contribute to the onset and characteristics of the disease.

Furthermore, VUSs are inevitable at the moment when using next-generation sequencing, and on average, they make up 40% of all variants for most diseases [60] and are even more common in Parkinson’s disease, as well as in other diseases where the concept of missing heritability exists [61]. VUS is a challenge in everyday clinical and research practice, and the problem can only be solved by further expanding knowledge in the field [60]. In this study, a significant number of VUSs were detected. Even though the difference was not statistically significant in our case, there are reports in the literature of a difference between the incidence of VUSs in groups with a positive family history versus other onset types of PD, possibly explained by the presence of yet unknown, extremely rare causative genes [62]. Overall, the percentage of VUSs in our study was similar to that in the literature, as reported by Trinh et al. [35] and others. It is also important to note the clinical problem of VUSs, given that patients have difficulty understanding this concept and often require genetic counseling before and after testing to explain it in an understandable way [63]. There are also significant differences in the reporting of VUSs between genetic laboratories and institutes, contributing to the current confusion in the field about what should and should not be reported to patients [64].

The study’s limitation is that not all patients underwent MLPA, which may limit the possibility of detecting copy number variants. Finally, we emphasize the importance of using Sanger sequencing to confirm GBA variants due to their similarity to the GBAP1 pseudogene.

## 5. Conclusions

The GIOPARK project represents the first effort to perform a comprehensive analysis of PD genetics in Croatia. We identified causative variants in 8.50% of patients, indicating the involvement of *GBA* variants in 4.58% of patients. We also found causative variants in the *PRKN*, *ITM2B*, *SNCA*, and *MAPT* genes. Also, a significant number of patients (25.49%) had variants of unknown significance (VUSs), highlighting the complexity of genetic interpretation. This underscores the need for further research to determine the clinical relevance of such variants. Our study also shows that even sporadic patients with Parkinson’s disease can have pathogenic variants, and due to the decreasing cost of genetic testing, there is no rationale for excluding such patients from testing.

## Figures and Tables

**Table 1 genes-15-00255-t001:** Sociodemographic data and variant percentages.

	Total		Early Onset (EO)		Familial Onset (FO)		Sporadic Onset (SO)	
Number of patients	*n* = 153		*n* = 42		*n* = 40		*n* = 71	
%	/		27.45%		26.14%		46.41%	
Sex	M	F	M	F	M	F	M	F
	*n* = 89	*n* = 64	*n* = 25	*n* = 17	*n* = 20	*n* = 20	*n* = 44	*n* = 27
%	58.17%	41.18%	59.52%	40.47%	50%	50%	61.97%	38.03%
Age of onset (years)	56.49 ± 12.61		42.07 ± 7.78		57.55 ± 12.03		63.8 ± 7.11	
Time to diagnosis (years)	1.52 ± 1.01		2.97 ± 0.83 *		1.04 ± 0.34 *		1.68 ± 0.98	
P/LP variants	*n* = 13		*n* = 6		*n* = 4		*n* = 3	
%	8.50%		13.95%		10%		4.23%	
VUS	*n* = 39		*n* = 12		*n* = 5		*n* = 22	
%	25.49%		28.57%		12.5%		30.98%	
B/LB variants	*n* = 20		*n* = 1		*n* = 5		*n* = 14	
%	13.16%		2.44%		12.5%		19.71%	

P—pathogenic variant, LP—likely pathogenic variant, VUS—variant of uncertain significance, LB—likely benign variant, B—benign variant, * *p* < 0.05.

**Table 2 genes-15-00255-t002:** Patients with pathogenic variants and variant characteristics.

Patient	Group	Age of Onset	Variants	Zygosity	GnomAD v4 AF	Applied ACMG Criteria	Classification
PT103	EO	36	GBA(NM_000157.4):c.1289C>T, p.Pro430Leu	HET	0.0002%	PS3, PS4, PM1, PM2	P
PT153	EO	32	GBA(NM_000157.4):c.882T>G, p.His294Gln	HET	0.0179%	PM2	VUS
			GBA(NM_000157.4):c.1226A>G, p.Asn409Ser	HET	0.1997%	PS3, PM3_VSTR, PP4	P
			GBA(NM_000157.4):c.1342G>C, p.Asp448His	HET	0.0112%	PS3, PM2, PM3_VSTR, PM5, PP4	P
PT104	EO	18	PRKN(NM_004562.3):c.823C>T, p.Arg275Trp	HOM	0.3009%	PS3, PM2, PM3_VSTR	P
PT105	EO	21	PRKN(NM_004562.3):c.823C>T, p.Arg275Trp	HOM	0.3009%	PS3, PM2, PM3_VSTR	P
PT009	EO	40	GBA(NM_000157.4):c.1448T>C, p.Leu483Pro	HET	0.0099%	PS3, PM3_VSTR, PM5, PM2, PP4	P
PT053	EO	42	PRKN(NM_004562.3):c.823C>T, p.Arg275Trp	HET	0.3009%	PS3, PM2, PM3_VSTR	P
PT028	F	52	GBA(NM_000157.4):c.1226A>G, p.Asn409Ser	HET	0.1997%	PS3, PM3_VSTR, PP4	P
PT112	F	60	GBA(NM_000157.4):c.1226A>G, p.Asn409Ser	HET	0.1997%	PS3, PM3_VSTR, PP4	P
PT038	F	67	SNCA(NM_000345.3):c.44T>C, p.Val15Ala	HET	0.0007%	PS3_SUP, PS4_MOD, PM2, PP1	LP
PT064	F	75	ITM2B(NM_021999.5):c.564+1G>T	HET	0.0002%	PVS1_STR, PM2	P
PT113	SO	58	GBA(NM_000157.4):c.1289C>T, p.Pro430Leu	HET	0.0002%	PS3, PS4, PM1, PM2	P
PT074	SO	57	GBA(NM_000157.4):c.1342G>C, p.Asp448His	HET	0.0112%	PS3, PM2, PM3_VSTR, PM5, PP4	P
PT039	SO	72	MAPT(NM_001377265.1):c.2263G>A, p.(Val755Ile)	HET	0.0025%	PS3_MOD, PS4	P

## Data Availability

The data presented in this study are available upon request from the corresponding author due to privacy and ethics concerns.

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
