# Peer review of "GiOPARK Project: The Genetic Study of Parkinson’s Disease in the Croatian Population"

_genes, 2024, doi:10.3390/genes15020255_

Round 1

Reviewer 1 Report

Comments and Suggestions for Authors

The authors report a survey of PD mutations in an European population. The research is well designed and conducted. There are some minor issues that must be solved before publication.

My main concern is about the statistics showed in the Results paragraph. 1) They say that "The FO patient group was the only one that did not have differences between genders" based in the distances to 50% in the other two groups. This is not exact since gender follows a binomial distribution that can differ from 50% in a certain quantity. The should modify this statement maybe giving confidence intervals. 2) For differences in time of disease diagnosis and age of onset they must use ANOVA test with post hoc analysis.

I would also like to see it the exome was done at low coverage (ex. 30x) or high coverage (ex. 100x).

The sentence "Even though long-read NGS can help mitigate this issue, it is still limited in recognizing recombinant alleles" is not exact since with long read NGS this issue can be solved.

Comments on the Quality of English Language

I found the quaiity of the English fair

Author Response

Thank you for taking the time to review our manuscript and for the comments. 

My main concern is about the statistics showed in the Results paragraph. 1) They say that "The FO patient group was the only one that did not have differences between genders" based in the distances to 50% in the other two groups. This is not exact since gender follows a binomial distribution that can differ from 50% in a certain quantity. The should modify this statement maybe giving confidence intervals. 2) For differences in time of disease diagnosis and age of onset they must use ANOVA test with post hoc analysis.

Answer 1: Thank you for the comment.
1) We have added the statistical significance of the binomial test in the results as follows: “89 men (58.17%) and 64 women (41,18%) patients participated in the research, with the difference being significant  (p=0.0084). The FO patient group did not have statistically significant differences between sexes (n=20 M, 50%, p=0.1253), which was observed in both SO (n=44 M, 61.97% p=0.0124) and EO (n=25 M, 59,25%, p=0.05).“
Following was added to the discussion: “The study cohort included more men than women, which is expected due to the higher risk of developing PD in males. The only exception was the FO group, although this could be due to the small sample size.“

2) We have added the data of the ANOVA test and Tukey HSD post-hoc test as follows: “There were also significant differences in the time of disease diagnosis in the cohort depending on the group (F= 25.229, p<0.00001). In the entire cohort, it was an average of 1.52 ± 1.01 years, while in the case of EO, it was significantly longer (2.97 ± 0.83, Q=7.19, p<0.05) and significantly shorter in the case of FO (1.04 ± 0.34, Q=5.18, p<0.05). In the case of SO, the time to diagnosis was similar to the entire group (1.68 ± 0.98, Q=1.00, p=0.892)“

Comment 2: I would also like to see it the exome was done at low coverage (ex. 30x) or high coverage (ex. 100x):

Answer 2: We thank the reviewer for the comment. We have included the information in the article. In short, we have achieved high coverage with the mean of mean on target coverage being 217x across the exomes included in the study.

Comment 3: The sentence "Even though long-read NGS can help mitigate this issue, it is still limited in recognizing recombinant alleles" is not exact since with long read NGS this issue can be solved.

Answer 3: We thank the reviewer for the observation, we agree and have removed the sentence from the manuscript.

Reviewer 2 Report

Comments and Suggestions for Authors

My suggestions:

1. In Table 2, I suggest adding a little more information. Authors may add, whether the mutation was observed in 1000Genomes and GnomAD database. PolyPhen2, SIFT, and CADD scores may be also added. 

2. I would also add a table on VUS variants with reference database frequencies and in silico predictions. I would also discuss these variants in Results section. 

3. There is a group of Early-onset PD patients. Do these patients have any family history?  

4. Were there any mutation detected in PD patients, which are associated with other neurodegenerative diseases? For example AD, FTD related mutations, CSF1R, NOTCH3 or PRNP mutations?

5. Were there any imaging done on the patients? Was there difference in the imaging data of PRKN and GBA gene carrier patients?  

6. Is it possible that VUS variants could play a role in PD onset? I think, the role of rare VUS variants in PD or other neurodegenerative diseases may not be ruled out. 

7. For the familial PD patients, was the segregation proven among patients? If they carried any pathogenic or probably pathogenic variants, could it be tested in affected relatives? 

8. MAPT gene may be involved in frontotemporal dementia (FTD) too. Was the MAPT mutation (Val698Ile) the authors found was observed in FTD patients? 

Comments on the Quality of English Language

English seems fine

Author Response

Thank you for taking the time to review our manuscript and for the comments.

  1. In Table 2, I suggest adding a little more information. Authors may add, whether the mutation was observed in 1000Genomes and GnomAD database. PolyPhen2, SIFT, and CADD scores may be also added. 

Answer 1: We thank the reviewer for the suggestion of adding additional information to table 2. To address this, we have added columns “Zygosity”, “GnomAD v4 AF (allele frequency)” and “Applied ACMG criteria”. We believe that especially the column including ACMG criteria will be valuable to readers, offering insight into the characteristics of the variants that establish their pathogenicity.

  1. I would also add a table on VUS variants with reference database frequencies and in silico predictions. I would also discuss these variants in Results section. 

Answer 2: We thank the reviewer for the observation regarding the importance of reporting variants of uncertain clinical significance. In our practice we adhere to the ACGS best practices guidelines for reporting of variants of uncertain clinical significance which includes the use of a temperature gradient sub-classification. We would not consider reporting variants of uncertain significance in the clinical setting, except “hot VUS” variants, for which no further evidence can be obtained at the time of interpretation. It is our opinion that the reasoning also applies in the research setting which we support with the following arguments:

  • Including a large number of VUS variants could significantly increase the volume of the presented data, detracting from the focus on clinically relevant findings which was the main objective of our study
  • Including a large number of VUS variants could lead to misinterpretation or overinterpretation of these findings in further publications.
  • Comprehensive comments on the large number of discovered VUS findings in the results section would furthermore distract from the clinically relevant findings representing the main focus of our study.

Given that no “hot VUS” variants have been discovered in the study it is our opinion that VUS variants should not be reported based on the above reasoning. Furthermore, we have assessed one VUS that could have met the criteria, but due to a new publication, it has been elevated to likely pathogenic and we have added it to the table. We thank the reviewer for the consideration of our arguments.

  1. There is a group of Early-onset PD patients. Do these patients have any family history?  

Answer 3: The patients in the early onset group did not have a positive family history. We have added the following to the manuscript methods to clarify: „Patients were divided into three groups based on the disease presentation: early onset (<50 years of age and no family history), familial onset (positive family history), and sporadic late onset (>50 years of age and no family history).“

  1. Were there any mutation detected in PD patients, which are associated with other neurodegenerative diseases? For example AD, FTD related mutations, CSF1R, NOTCH3 or PRNP mutations?

Answer 4: We thank the reviewer for bringing up an interesting topic. Apart from what has been described in the article, no genetic variants associated with other neurodegenerative disorders have been detected.

  1. Were there any imaging done on the patients? Was there difference in the imaging data of PRKN and GBA gene carrier patients?  

Answer 5: All patients underwent MRI as routine workup for Parkinson's disease. There were differences as PRKN patients had more white matter abnormalities, and in one patient firstly decribed as a possible demyelinating disorder prior to the onset of parkinsonism. Although these differences are interesting, we belive they are out of scope for the manuscript.
Some of the patients from this study are a part of the longitudinal prospective research arm of the GiOPARK project, which will in one part focus on the MRI characteristics, but has not yet been completed.

  1. Is it possible that VUS variants could play a role in PD onset? I think, the role of rare VUS variants in PD or other neurodegenerative diseases may not be ruled out. 

Answer 6: We thank the reviewer for the observation. It is our opinion that variants of uncertain significance should be treated very carefully due to the reasoning presented under answer number 2 of this reply. We would again like to thank the reviewer for consideration of our arguments.

  1. For the familial PD patients, was the segregation proven among patients? If they carried any pathogenic or probably pathogenic variants, could it be tested in affected relatives? 

Answer 7: Unfortunately we could not do the segregation analysis as the affected relatives were deceased in all four patients.

  1. MAPT gene may be involved in frontotemporal dementia (FTD) too. Was the MAPT mutation (Val698Ile) the authors found was observed in FTD patients? 

Answer 8: The variant p.Val698Ile in MAPT gene detected in one patient in our study has been previously reported in patients with FTD and corticobasal syndrome. Our patient does not have a clinical presentation consistent with FTD, or atypical parkinsonism, even though there is a present cognitive decline and a borderline phosphor-tau/tau ratio in cerebrospinal fluid.

Round 2

Reviewer 2 Report

Comments and Suggestions for Authors

Authors fulfilled my suggestions, thank you.